# Active-Passive SimStereo – Benchmarking the Cross-Generalization Capabilities of Deep Learning-based Stereo Methods

**Laurent Jospin**[1]* **Allen Antony**[1] **Lian Xu**[1] **Hamid Laga**[2] **Farid Boussaid**[1]
**Mohammed Bennamoun**[1]
[1]University of Western Australia  [2]Murdoch University
{laurent.jospin,lian.xu,farid.boussaid,mohammed.bennamoun}@uwa.edu.au
H.Laga@murdoch.edu.au

## Abstract

In stereo vision, self-similar or bland regions can make it difficult to match patches between two images. Active stereo-based methods mitigate this problem by projecting a pseudo-random pattern on the scene so that each patch of an image pair can be identified without ambiguity. However, the projected pattern significantly alters the appearance of the image. If this pattern acts as a form of adversarial noise, it could negatively impact the performance of deep learning-based methods, which are now the de-facto standard for dense stereo vision. In this paper, we propose the Active-Passive SimStereo dataset and a corresponding benchmark to evaluate the performance gap between passive and active stereo images for stereo matching algorithms. Using the proposed benchmark and an additional ablation study, we show that the feature extraction and matching modules of a selection of twenty selected deep learning-based stereo matching methods generalize to active stereo without a problem. However, the disparity refinement modules of three of the twenty architectures (ACVNet, CascadeStereo, and StereoNet) are negatively affected by the active stereo patterns due to their reliance on the appearance of the input images.

## 1 Introduction

Stereo vision is used by many artificial or natural vision systems to acquire depth information from a pair of 2D projective views of the 3D world. In the context of computer vision, stereo matching operates in a multi-step pipeline (Fig. 2) composed of: (**i**) a feature volume construction from the left and right views, (**ii**) a cost volume computation, which may be coupled with a regularization module, (**iii**) a disparity extraction from the cost volume, which is done using the argmin function, and (**iv**) a disparity refinement module, which may also use the cost volume and/or the image features as additional cues. The central step in this pipeline is the construction of the cost volume, which is a function $C(x, y, d)$ that measures how unlikely a pixel of spatial coordinates $(x, y)$ is to be assigned a disparity value $d$. Textureless and repetitive patterns in images can produce flat or periodic cost curves in the cost volume, leading to erroneous disparity maps in passive stereo systems, where only a pair of cameras is used. To address this issue, active stereo-based methods [13] project a pseudo-random light pattern on the scene to remove the textureless or self-similar areas in the stereo images (Fig. 1). Active stereo is now a critical component in many applications such as augmented reality [22] and robotics [3]. They are also part of consumer electronics devices such as smartphones [26].

Traditional stereo matching pipelines rely solely on closed-form formulations [29]. However, in recent years, learning-based methods have led to a series of breakthroughs in the field. Early learning-

36th Conference on Neural Information Processing Systems (NeurIPS 2022) Track on Datasets and Benchmarks.

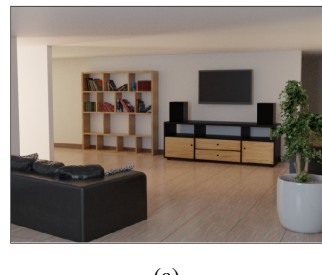
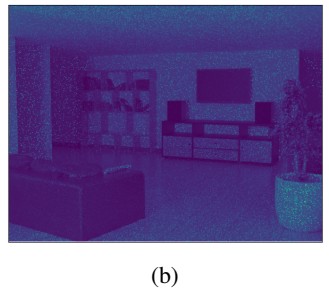
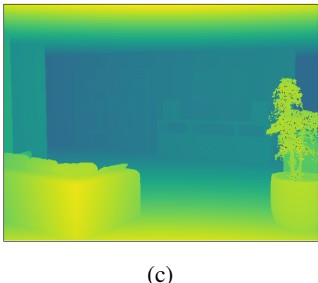

| (a) | (b) | (c) |

Figure 1: A sample from our dataset, with realistic (a) passive and (b) active stereo images along with (c) their corresponding perfect ground truth disparities. The proposed dataset allows the comparison of the relative performance of stereo vision methods when used for passive or active stereo matching.

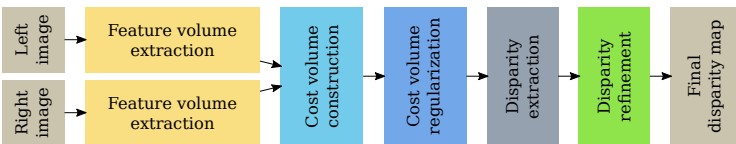

Figure 2: The typical stereo matching pipeline.

based methods focused on replacing one or more blocks in the traditional pipeline with a deep neural network. The latest methods, however, address the problem in an end-to-end fashion; see [15] for a detailed survey. Due to the lack of public active stereo datasets and the fact that passive stereo was perceived as more challenging, most of these models have been trained for the passive stereo problem. An important property of the closed-form formulae used in traditional stereo matching methods is that when non-self similar texture is added to the scene, their performance monotonically increases. This key feature is at the core of active stereo systems [21]. If one can determine that the latest deep learning methods can also leverage active pseudo random noise to improve their prediction, this would show that these methods are indeed learning to match similar regions of the images rather than fitting some bias into the data. Additionally, it provides some insight into the models' generalization ability, which is important for their safe deployment in their intended application (e.g., autonomous driving).

Under ideal conditions, deep learning-based methods are expected to behave in a similar fashion to their non learning counterpart and exhibit improved performance when additional pseudo-random texture is added to the scene. Yet, many large-scale deep learning models see a degradation of their performances when used on datasets that are only slightly different from their original training datasets [43]. They often require an adaptation procedure to generalize to new unseen domains [33]. Furthermore, they can be severely affected by even little adversarial noise under certain circumstances [32], as they are prone to overfitting on small biases present in their training data [23]. However, these flaws are not everywhere. For example, it has already been shown that once simulated images are close enough to real images, deep learning stereo systems generalize without issues [34]. Also, unlike adversarial noise, the pseudo random patterns used in active stereo have not been learned specifically to cause failure for deep learning models. This means that existing deep learning methods might generalize to the active stereo domain without any form of fine-tuning.

In this work, we investigate how different state-of-the-art deep learning-based stereo matching architectures are impacted when presented with active, instead of passive, stereo images. To make the evaluation of the generalization ability of stereo vision models easier, we propose Active-Passive SimStereo, a novel dedicated dataset composed of computer-generated images rendered using a physically-based rendering engine. The proposed dataset provides both active and passive frames for each given scene. This allows to evaluate and compare the performance of each algorithm on active and passive stereo using exactly the same scenes. The data set is publicly available at `https://dx.doi.org/10.21227/gf1e-t452`.

The remaining parts of the paper are organized as follows. Section 2 reviews the related work. Section 3 describes the proposed dataset. Section 4 presents the proposed benchmark used for

evaluation. Section 5 presents and discusses the results of existing methods. Finally, Section 8 concludes the paper.

## 2   Related Work

Many datasets and benchmarks have been proposed for passive stereo vision including the popular Middlebury dataset [29, 11], whose latest version uses a precise but expensive reconstruction pipeline to acquire the ground truth [30]. The corresponding Middlebury Stereo Evaluation benchmark is widely used to evaluate stereo vision algorithms. Due to the challenges associated with the 3D ground-truth acquisition, the aforementioned dataset only contains a small amount of labelled data, which is not sufficient to train large-scale deep architectures. Subsequently, the Scene Flow datasets have been proposed [19]. They contain a large number of simulated image pairs with ground-truth optical flows and disparities generated from open source motion graphics short movies or randomized virtual 3D objects. However, the appearance of these simulated scenes is not realistic. Thus, most deep learning-based models for stereo vision need to be fine-tuned after being trained on the Scene Flow datasets. The UnrealStereo4K simulated dataset [34] was later proposed to provide higher resolution and more realistic images, taken from video games scenes.

One of the most popular applications of stereo vision is autonomous driving, since vision based systems offer a cost-effective alternative or complement to LIDAR-based systems for depth measurement. Thus, many datasets and benchmarks have been specifically developed for this application. Examples include the KITTI Vision suite [20, 7], which is currently the most popular stereo vision benchmark for autonomous driving, DrivingStereo [40], which is a large dataset commonly used for training rather than evaluation, and ApolloScape [12], which provides a benchmark suite for different challenges related to autonomous driving, including stereo vision. The ground truth of these datasets was obtained using a LIDAR-based system. Occasionally, a recognition system was also used to detect and categorize cars in images before aligning a CAD model onto the LIDAR depth map [20, 12]. The inherent noise associated with these various processing steps implies that the ground truth cannot be trusted for very precise reconstructions. However, given that autonomous driving scenarios do accommodate a disparity error of one or two pixels, this is not a problem for the intended use of those datasets.

For active stereo vision, there are far fewer public datasets, none of which has become popular for training or evaluating deep learning-based stereo matching methods. The few end-to-end methods trained for active stereo use soft labels, i.e., labels with associated uncertainty, such as the depth generated by stereo cameras [44, 41]. Other methods did also use self-supervision, e.g., by using the information conserved when compressing a given image patch as supervisory signal [28]. Simulation techniques have also been proposed to generate semi-realistic images from CAD models [25] based on screen space projection of texture. This approach has been used on multiple occasions [28, 44], but none of the produced datasets has been made public. In this work, We use a similar approach but with a physically-based rendering pipeline to improve the realism of the scenes, making our dataset more suitable for evaluation.

Datasets providing images for both active and passive stereo matching are even more scarce. To the best of our knowledge, only UnrealStereo4K [34] has monocular active frames for a subset of the images, but this part of the dataset has not been made publicly available. Furthermore, monocular active depth estimation is a slightly different problem from active stereo vision [27], as the matching is performed between a pattern and an image, rather than between two images with a projected pattern. Thus, there exists no public dataset available for evaluating stereo models on active stereo images or evaluating the generalization capabilities of these models.

## 3   The ACTIVE-PASSIVE SIMSTEREO Dataset

Simulation offers both benefits and challenges for dataset creation. The size of real stereo datasets with high quality ground truth like Middlebury 2014 is limited because of the complex setups and amount of work needed [30]. On the other hand, automated pipelines like the ones used for Kitti [20] are noisy. The Kitti benchmark is, therefore, limited to $BAD_N$ metrics (see Section 4) with large $N$ and is not suitable for subpixel accuracy comparisons. Simulation on the other hand makes labelling cheaper and noiseless. It also allows to generate the exact same image twice, once for

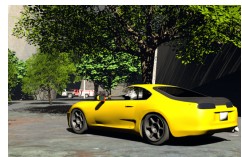 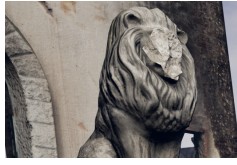 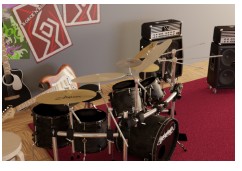 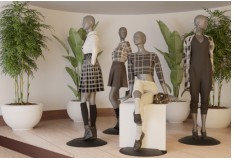

(a) Sceneflow [19]  (b) UnrealStereo4K [34]  (c) Ours  (d) Ours

Figure 3: Illustration of the benefits of a physically-based rendering pipeline. In our images (c), (d), indirect lighting creates soft and realist shadows and specularities, which is not the case with existing simulated datasets (a), (b).

active stereo and once for passive stereo. This greatly reduces biases when measuring the generalization ability of a given model, since the geometry does not change between frames. However, the simulation procedure may add a domain gap with real images. In computer vision, simulated data can be obtained using computer generated imagery (CGI). In recent years, CGI software has made tremendous advances. Using advanced CGI techniques, generating synthetic scenes that are indistinguishable from real images and match quite closely the acquisition process of real cameras became possible [8]. This is done by using physically-based rendering (PBR), i.e., a rendering engine simulating the physical behaviour of light [24]. Figure 3 compares images from our dataset with images from other simulated datasets [34, 19], which used real-time render instead of PBR. The most visible benefit of using PBR is that it produces more accurate indirect lighting, thereby creating softer and more accurate shadows. A better rendering of non-Lambertian materials [24] is another benefit. This significantly reduces the domain gap between real and simulated data once a synthetic scene is created. In this paper, we purchased a series of high-quality realistic 3D assets to create realistic scenes for our dataset. Our dataset is not intended for training but for performance evaluation and fine-tuning, so we prioritize quality and diversity over quantity. To increase the number and the diversity of scenes, we also generated images that contain procedurally-generated shapes and images with abstract objects.

### 3.1 Simulation Procedure

For each 3D scene, we designed two lighting setups. The first corresponds to a passive stereo acquisition scenario without any pseudo-random pattern while the second one corresponds to an active stereo acquisition scenario with a collection of lights projecting a pseudo-random pattern, see Figure 4). We used the Cycles path tracing engine integrated into Blender [4] for rendering. We used standard shaders for the non-textured light sources. For the pseudo-random pattern light projectors, we used a custom programmed shader to generate a pattern resembling the one from the RealSense cameras. The light intensity $I$, is a function of the incoming direction $d$:

$$I(\boldsymbol{d}) = p\left((1 - W_{s_1}(\boldsymbol{d} + \boldsymbol{t})^2)^{p_1} + c\right)\left(1 - W_{s_2}(\boldsymbol{d} + \boldsymbol{t})^2\right)^{p_2}. \tag{1}$$

Here, $W_{s_1}$ and $W_{s_2}$ are Whorley noise patterns [36]; $s_1$ and $s_2$ are the scale factors of their respective Whorley noise pattern with $s_1 < s_2$; $p_1$ and $p_2$ are two light intensity correction factors with $p_1 \ll p_2$; $c$ is the minimal power of the $W_{s_2}$ pattern; $\boldsymbol{t}$ is a random translation of the texture space to generate different patterns for different lights; and $p$ is the power gain of the lamp, expressed in Watts.

The ground truth is then extracted from the depth pass $z$ generated by the rendering software. $z$ measures the distance between the visible point in the 3D scene and the optical center of the camera. The ground truth disparity $\check{d}$ can be computed as:

$$\check{d} = \frac{Bf}{z}, \tag{2}$$

where $B$ is the stereo camera baseline and $f$ is the camera focal length in pixel. We used a Baseline of 0.16m (where m here refers to the unit used internally by Blender, not the actual meters in the real world), and cameras with a focal length of $48.61$mm, which amounts to $888.89$ pixels at standard resolution for a 35mm film equivalent sensor.

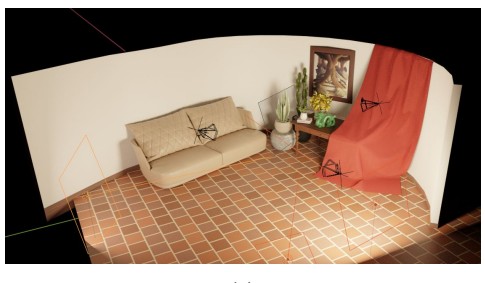
(a)

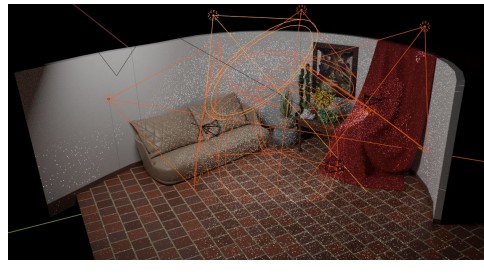
(b)

Figure 4: The two different light setups to simulate (a) passive and (b) active stereo acquisition in a given 3D scene.

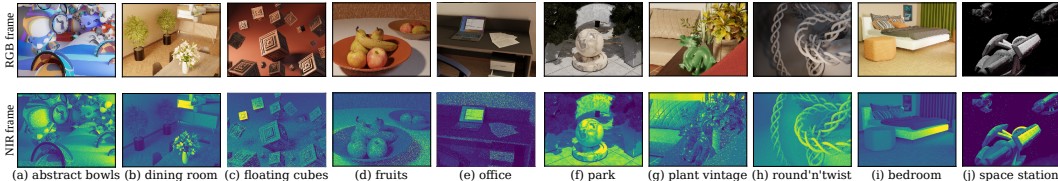

(a) abstract bowls (b) dining room (c) floating cubes (d) fruits (e) office (f) park (g) plant vintage (h) round'n'twist (i) bedroom (j) space station

Figure 5: Sample images from the proposed test set.

## 3.2 The Dataset

The proposed dataset contains $515$ image pairs split into a training set ($80\%$ of the images) and a test set ($20\%$ of the images) used for benchmarking. The test set contains $103$ image pairs (Fig. 5), comprising different shapes (e.g., large flat surfaces such as floors or small areas with depth discontinuities such as plant leaves), depth ranges, and styles (i.e., realistic scenes or abstract compositions). The remaining $412$ image pairs are contributed as a training set. We use the standard resolution ($640 \times 480$ pixels) for the benchmark as it matches or approaches the resolution of most stereo cameras. It also guarantees that the images can be processed by most methods even on memory-constrained hardware. Despite the small number of images, the test set is shown to be large enough to evaluate deep learning methods. Detailed experiments to demonstrate this are provided in the Supplementary Material. Our dataset is also large enough for fine-tuning (see Section 7).

In addition to the benchmark, we provide a simulation Blender file with the specific shader, as well as the python code to post-process the images.

Table 1: Comparison of the resolution and disparity ranges of different datasets.

| Dataset | Generation method | # images pairs | Resolution [px] | Disparity [px] Min | Mean | Max |
|---|---|---|---|---|---|---|
| Ours (train) | Ray tracing rendering | 412 | $640 \times 480$ | 0.00 | 21.77 | 212.67 |
| Ours (test) | Ray tracing rendering | 103 | $640 \times 480$ | 0.00 | 25.12 | 129.67 |
| Middlebury 2014 [30] | Large baseline active stereo | 33 | $2850 \times 1900$[1] | 28.94 | 148.36 | 695.61 |
| Kitti 2012 [7] | Real images + Laser scanner | 389 | $1240 \times 380$[1] | 4.11 | 38.32 | 227.99 |
| Kitti 2015 [20] | Real images + Laser scanner + 3D cad object alignment | 400 | $1240 \times 380$[1] | 4.46 | 33.62 | 229.96 |
| Sceneflow [19] | Screen space rasterization | 39049 | $960 \times 540$ | 1.12 | 39.87 | 940.75 |
| UnrealStereo4k [34] | Screen space rasterization | 7200 | $3840 \times 2160$ | 0.01 | 173.20 | 1515.60 |

[1] The images in these datasets have variable sizes. Thus, the values given here are an approximation.

## 4 The Benchmark

We use five different scores to compare the generalisation abilities of different methods. The most important metric in stereo vision is the BAD-N, which measures the proportion of pixels above a given error threshold $N$. It is computed over a test set $T$ as:

$$BAD_N = \frac{1}{\|T\|} \sum_{I \in T} \frac{\sum_{i=0}^{h_I} \sum_{j=0}^{w_I} \mathbf{1}_{|\Delta d_{i,j}| > N}}{hw}, \tag{3}$$

where $h$ and $w$, respectively, are the height and the width of the image $I$, $\mathbf{1}_A$ is the indicator function of $A$ and $\Delta d$ is the disparity error for the image $I$. We report results for $N \in \{0.5, 1, 2, 4\}$. Lower $BAD_N$ scores indicate a better accuracy in reconstructing the disparity. In the rest of the paper, we will indicate this by appending a down arrow ($\downarrow$) after the name of these metrics.

We also use the Mean Absolute Error (MAE) and the Root Mean Square Error (RMSE). The former is a good estimate of the expected amplitude of the error of a given method, and the latter, especially when compared to the MAE, is a good indicator of the presence of outlier points with large errors. The RMSE and MAE scores over $T$ are computed as:

$$RMSE_T = \frac{1}{\|T\|} \sum_{I \in T} \sqrt{\frac{\sum_{i=0}^{h} \sum_{j=0}^{w} \Delta d_{i,j}^2}{hw}}, \quad MAE_T = \frac{1}{\|T\|} \sum_{I \in T} \frac{\sum_{i=0}^{h} \sum_{j=0}^{w} |\Delta d_{i,j}|}{hw} \ . \quad (4)$$

These scores are the average of the corresponding metric over the test set. Lower MAE and RMSE indicate a better accuracy in reconstructing the disparity. In the rest of the paper, we will indicate this by appending a down arrow ($\downarrow$) after the name of these metrics.

A consequence of using the MAE and RMSE is that if a method exhibits low performances on a specific image, the overall score of the method will be greatly influenced by this single image. For an absolute performance evaluation benchmark, this is not a problem. However, this is an issue when evaluating the relative performance variation resulting from a domain change (e.g., from passive stereo to active stereo). To mitigate this use, we measure the mean relative score variation across all images. Given a metric $M$, the relative score variation $R_M$ is computed as:

$$R_M = \frac{1}{\|T\|} \sum_{I \in T} \frac{M_{I_P} - M_{I_A}}{M_{I_P}} \ , \quad (5)$$

where $M_{I_P}$ and $M_{I_A}$ are the metric scores evaluated on the passive stereo results and active stereo results of an image $I$, respectively. In this paper, we focus on $R_{MAE}$ and $R_{BAD_2}$.

We also report the proportion $P_M$ of the testing images in which the active stereo results outperform their passive stereo counterparts in terms of the metric $M$. It is formulated as:

$$P_M = \frac{1}{\|T\|} \sum_{I \in T} \mathbf{1}_{M_{I_A} < M_{I_P}}. \quad (6)$$

In this paper, we focus on $P_{MAE}$ and $P_{BAD_2}$. We report in the supplementary material other variants of those metrics.

Higher P and R scores indicate a better generalization of the method from passive stereo to active stereo. In the rest of the paper, we will indicate this by appending an up arrow ($\uparrow$) after the name of these metrics.

Finally, discontinuities in the depth image are an important parameter in evaluating stereo reconstruction methods. Discussing the detailed performances of existing methods would be beyond the scope of this paper, which focuses on the generalization capabilities from passive to active stereo and vice versa. Nonetheless, we also included, in the supplementary material, versions of all the metrics presented above, restricted either on the edge regions of the image or on its flat regions.

## 5  Results on Existing Methods

We used the proposed benchmark to evaluate state-of-the-art, end-to-end deep neural networks for stereo-matching. We considered 20 methods for which the source code and the pre-trained models are available. In the supplementary material, we also evaluate a selection of traditional non-learning methods. Those methods are listed in Table 2, along with the datasets used to train the models. We used mainly the models weights trained on SceneFlow [19] and fine-tuned on KITTI 2015 [20] as this approach is the standard for deep stereo models [15]. For each method, we list the reported D1 score in the KITTI 2015 benchmark [20]. This is the proportion of pixels with an error greater than 3px or 5% of the disparity. We also report, if available, the $BAD_2$ and MAE scores of the method in the Middlebury benchmark [30].

Table 2: Evaluated methods with training sets and results on public benchmarks

| Method | Stereo type | Training for our evaluation | | KITTI 2015 [20] | Middlebury [30] | |
| | | Train set | Fine-tuned set | D1-all ↓ | BAD$_2$ ↓ | MAE ↓ |
|---|---|---|---|---|---|---|
| AANet [38] | Passive | SceneFlow | KITTI 2015 | 2.55% | 25.20% | 8.88px |
| ACVNet [37] | Passive | SceneFlow | KITTI 2015 | 1.65% | 13.60% | 8.24px |
| AnyNet [35] | Passive | SceneFlow | KITTI 2015 | 6.20% | - | - |
| CascadeStereo [9] | Passive | SceneFlow | KITTI 2015 | 2.00% | 18.80% | 4.50px |
| CREStereo [16] | Passive | CREStereo | ETH3D | 1.69% | 3.71% | 1.15px |
| Deep-Pruner (best) [6] | Passive | SceneFlow | KITTI 2015 | 2.15% | 30.10% | 4.80px |
| Deep-Pruner (fast) [6] | Passive | SceneFlow | KITTI 2015 | 2.59% | - | - |
| GA-Net [42] | Passive | SceneFlow | KITTI 2015 | 2.59% | 18.90% | 12.20px |
| GwcNet [10] | Passive | SceneFlow | KITTI 2015 | 2.11% | - | - |
| HighResStereo [39] | Passive | SceneFlow | KITTI 2015 | 2.14% | 10.20% | 2.07px |
| Lac-GwcNet [18] | Passive | SceneFlow | KITTI 2015 | 1.77% | - | - |
| MobileStereoNet3D [31] | Passive | SceneFlow | KITTI 2015 | 2.10% | - | - |
| MobileStereoNet2D [31] | Passive | SceneFlow | KITTI 2015 | - | - | - |
| PSM-Net [1] | Passive | SceneFlow | KITTI 2015 | 2.32% | 42.10% | 6.68px |
| RAFT-Stereo [17] | Passive | SceneFlow | - | - | 4.74% | 1.27px |
| RealTimeStereo [2] | Passive | SceneFlow | KITTI 2015 | 7.54% | - | - |
| SMD-Nets [34] | Passive | UnrealStereo4K | KITTI 2015 | 2.08% | - | - |
| SRH-Net [5] | Passive | SceneFlow | KITTI 2015 | - | - | - |
| StereoNet [14] | Passive | SceneFlow | - | - | - | - |
| ActiveStereoNet [44] | Active | Self-supervised | - | - | - | - |

*As we are more interested in the generalization abilities of the different methods than their peak performances, we did not fine-tune any of these methods to our dataset.* This is to ensure the measured generalization performances are not biased by fine-tuning for one specific modality in our dataset. On the other hand, fine-tuning for both modalities would not give the same insights on the generalization abilities of each method. However, this means that the reported performances should not be considered as the best achievable performances of the studied methods. In a second step, we fine-tune the methods which failed to generalize on our dataset to check if it can be used to fine-tune passive stereo methods on active stereo (Section 7).

In this paper, we focus on the main aggregate results as well as on our general observations when analysing the error maps. The detailed scores per image are provided in the supplementary material.

Table 3: Evaluation results of state-of-the-art methods on passive and then active stereo images.

| Method | Passive stereo images | | | | | | Active stereo images | | | | | |
| | RMSE$_T$ ↓ | MAE$_T$ ↓ | BAD$_{0.5}$ ↓ | BAD$_1$ ↓ | BAD$_2$ ↓ | BAD$_4$ ↓ | RMSE$_T$ ↓ | MAE$_T$ ↓ | BAD$_{0.5}$ ↓ | BAD$_1$ ↓ | BAD$_2$ ↓ | BAD$_4$ ↓ |
|---|---|---|---|---|---|---|---|---|---|---|---|---|
| AANet [38] | 8.35px | 4.07px | 66% | 45% | 30% | 20% | 6.73px | 2.02px | 43% | 20% | 12% | 8% |
| ACVNet [37] | 9.22px | 4.00px | 36% | 25% | 17% | 12% | 3.49px | 1.31px | 23% | 13% | 8% | 5% |
| AnyNet [35] | 8.80px | 5.31px | 84% | 69% | 46% | 29% | 6.13px | 3.43px | 80% | 61% | 32% | 18% |
| Cascade-Stereo [9] | 10.12px | 4.95px | 66% | 39% | 23% | 16% | 4.48px | 2.07px | 64% | 33% | 14% | 8% |
| CREStereo [16] | 1.75px | 0.71px | 21% | 14% | 8% | 4% | 1.44px | 0.32px | 7% | 4% | 2% | 1% |
| Deep-Pruner (best) [6] | 7.07px | 3.80px | 50% | 34% | 24% | 17% | 3.16px | 1.12px | 23% | 12% | 7% | 5% |
| Deep-Pruner (fast) [6] | 7.97px | 4.73px | 67% | 47% | 32% | 22% | 4.27px | 1.91px | 41% | 23% | 13% | 9% |
| GA-Net [42] | 7.89px | 4.01px | 59% | 39% | 27% | 19% | 5.05px | 1.56px | 27% | 14% | 9% | 7% |
| GwcNet [10] | 10.38px | 5.17px | 74% | 50% | 33% | 22% | 4.41px | 1.80px | 53% | 24% | 11% | 8% |
| High-Res-Stereo [39] | 4.87px | 2.94px | 54% | 37% | 24% | 16% | 3.11px | 1.30px | 38% | 20% | 10% | 5% |
| Lac-GwcNet [18] | 7.71px | 3.95px | 58% | 39% | 26% | 18% | 3.34px | 1.25px | 31% | 14% | 8% | 5% |
| MobileStereoNet3D [31] | 9.81px | 5.06px | 81% | 61% | 38% | 23% | 4.44px | 2.11px | 70% | 38% | 14% | 8% |
| MobileStereoNet2D [31] | 7.68px | 4.49px | 79% | 59% | 36% | 23% | 4.37px | 1.88px | 59% | 28% | 13% | 8% |
| PSM-Net [1] | 6.57px | 4.23px | 93% | 82% | 45% | 21% | 3.94px | 2.32px | 93% | 80% | 27% | 7% |
| RAFT-Stereo [17] | 2.20px | 0.92px | 27% | 17% | 10% | 5% | 1.68px | 0.47px | 13% | 7% | 3% | 2% |
| RealTimeStereo [2] | 7.69px | 4.71px | 80% | 64% | 44% | 28% | 5.30px | 2.87px | 72% | 50% | 28% | 15% |
| SMD-Nets [34] | 12.32px | 6.48px | 77% | 58% | 40% | 27% | 5.26px | 2.24px | 63% | 34% | 15% | 9% |
| SRH-Net [5] | 7.43px | 4.24px | 76% | 49% | 29% | 19% | 3.95px | 1.59px | 57% | 21% | 9% | 5% |
| StereoNet [14] | 10.98px | 4.11px | 55% | 37% | 26% | 18% | 7.74px | 1.79px | 44% | 23% | 12% | 7% |
| ActiveStereoNet [44] | 21.57px | 9.39px | 60% | 46% | 35% | 28% | 6.92px | 2.32px | 36% | 22% | 14% | 9% |

Table 3 reports results on passive and active stereo images. We observed that, when evaluated on active stereo images, all considered methods show an improvement in their performance for all considered metrics. *Please keep in mind that not all methods have been trained on the same dataset or even domain; our benchmark aims to evaluate the relative performances of the different methods rather than their absolute performances.* ActiveStereoNet [44] stands out as the worst performing method for passive stereo but its performance drastically improves when presented with active stereo images. This is due to the fact that it is the only one trained on active stereo images. A model trained for active stereo is not expected to generalize well on passive stereo without adaptation since the domain shift makes matching much harder. ActiveStereoNet is far from being the best performing

Table 4: Relative scores (equations 5 and 6) for the state-of-the-art methods.

| Method | $P_{MAE} \uparrow$ | $P_{BAD_2} \uparrow$ | $R_{MAE} \uparrow$ | $R_{BAD_2} \uparrow$ |
|---|---|---|---|---|
| AANet [38] | 87% | 99% | 35% | 55% |
| ACVNet [37] | 71% | 74% | 30% | 37% |
| AnyNet [35] | 93% | 98% | 29% | 29% |
| Cascade-Stereo [9] | 65% | 71% | 25% | 28% |
| CREStereo [16] | 80% | 62% | 41% | 34% |
| Deep-Pruner (best) [6] | 94% | 100% | 52% | 62% |
| Deep-Pruner (fast) [6] | 95% | 98% | 47% | 56% |
| GA-Net [42] | 87% | 96% | 43% | 59% |
| GwcNet [10] | 99% | 99% | 55% | 63% |
| High-Res-Stereo [39] | 84% | 85% | 36% | 49% |
| Lac-GwcNet [18] | 96% | 96% | 55% | 63% |
| MobileStereoNet3D [31] | 96% | 99% | 45% | 59% |
| MobileStereoNet2D [31] | 96% | 99% | 52% | 65% |
| PSM-Net [1] | 90% | 93% | 30% | 38% |
| RAFT-Stereo [17] | 81% | 73% | 37% | 39% |
| RealTimeStereo [2] | 92% | 97% | 34% | 35% |
| SMD-Nets [34] | 95% | 97% | 54% | 60% |
| SRH-Net [5] | 97% | 98% | 48% | 64% |
| StereoNet [14] | 84% | 85% | 38% | 45% |
| ActiveStereoNet [44] | 99% | 99% | 68% | 58% |

model on active stereo images despite being the only one trained on that domain. The neural network architecture is probably at play here. ActiveStereoNet uses the same architecture as StereoNet [14], albeit a different training method. StereoNet is not the best performing method on passive stereo and is outperformed by ActiveStereoNet on active stereo images, which demonstrates the benefits of the training strategy proposed by the authors of ActiveStereoNet.

The relative score improvements, reported in Table 4, also show that, overall, existing methods have a good capability to generalize to active stereo. ACVNet [37], Cascade Stereo [9] as well as CREStereo [16] and RAFT-Stereo [17] are the only methods that seem to have issues generalizing, as their $P_{MAE}$ and $P_{BAD_2}$ scores are way lower than the other methods. For CREStereo [16] and RAFT-Stereo [17], it turns out this is more due to the fact that these methods perform so well on passive stereo, see Table 3, that they have less room for improvement when moving to active stereo. For ACVNet [37] and Cascade Stereo [9], this is because moving to the active stereo domain causes artifacts in the reconstructed disparities, see Section 6 for more details.

These results are encouraging and show that, when trained on passive stereo, current state-of-the-art deep learning methods are able to generalize to active stereo. Generalizing to passive stereo for a method trained on active stereo, seems to be much harder. However, the results are not very conclusive, since only one method is covered, and it has been trained in a self-supervised fashion.

The details of these results, as well as the performance of each method on each image, are provided in the results Excel sheet included in the supplementary material.

## 6 Ablation study

The visual inspection of the results shows that many visual artifacts are difficult to capture by aggregate metrics; see for example Figure 7). The refinement module at the end of the stereo matching pipeline is the one most likely to be impacted by a switch from passive to active stereo. Consequently, we chose to test all methods a second time, but with their final refinement module deactivated. Figure 6 shows error maps for methods which did generalize to active stereo while Figure 7 shows error maps of the methods which had issues generalizing (ACVNet [37], Cascade Stereo [9] and StereoNet [14]).

Looking into more details at the error maps of Fig. 7, we can notice different effects that may explain these observations. In certain rare situations, artifacts will appear in the depth map reconstructed using active stereo; see Fig. 7a and Fig. 7b. This indicates that under specific circumstances, the pseudo random pattern used for active stereo can act like an adversarial noise. ACVNet has such artifacts around small objects edges, like for example the handle of the chair in Fig. 7a. ACVNet uses

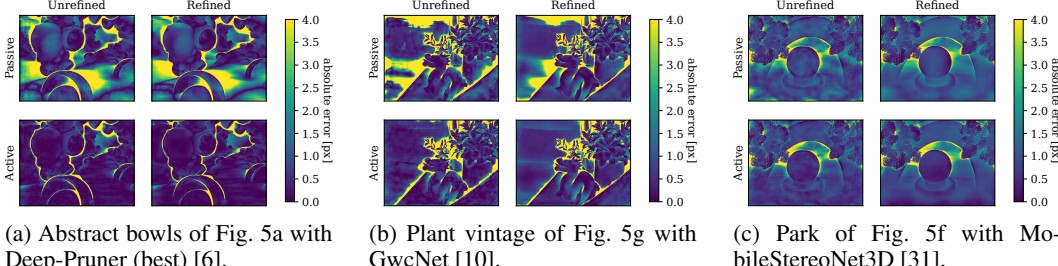

(a) Abstract bowls of Fig. 5a with Deep-Pruner (best) [6].

(b) Plant vintage of Fig. 5g with GwcNet [10].

(c) Park of Fig. 5f with MobileStereoNet3D [31].

Figure 6: Detailed error maps comparison between the unrefined and refined disparity maps for methods without apparent problems in active stereo compared to passive stereo.

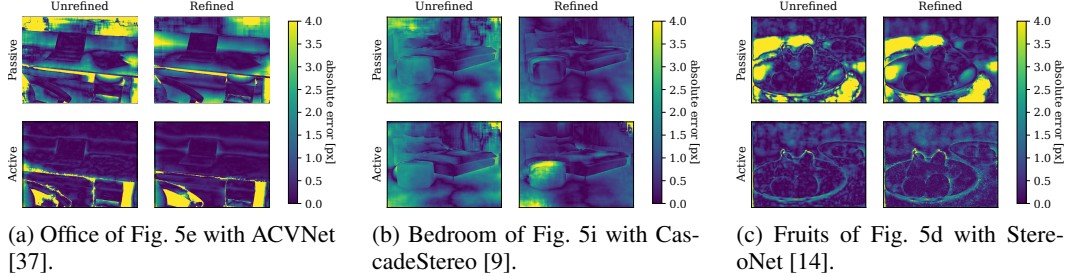

(a) Office of Fig. 5e with ACVNet [37].

(b) Bedroom of Fig. 5i with CascadeStereo [9].

(c) Fruits of Fig. 5d with StereoNet [14].

Figure 7: Detailed error maps comparison shows some artifacts not present in the unrefined active disparity appearing in the refined active disparity of certain methods, showing that their refinement module is negatively impacted by the active stereo pseudo random pattern.

an attention based mechanism to post-process the matching cost volume. This tends to generate edge artifacts in active stereo images. The refinement process is negatively impacted by these artifacts, which further degrade the quality of the final disparity map.

Cascade Stereo has some of the most visible examples of such effects, where large error patches seem to appear in the active stereo disparity map, while they are not present in the passive disparity map. If refinement is turned off, the error maps for active and passive stereo are quite similar; see Fig. 7b. Note that the errors between unrefined and refined disparities can vary quite a lot for CascadeStereo compared to the other methods in Figure 6 and Figure 7. This shows that CascadeStereo relies extensively on its final refinement module to improve its results, especially compared to other methods. This, in turn, explains why it does not generalize as well as other architectures; see Table 4.

StereoNet also exhibits a strange behaviour; see Fig. 7c. While the network is able to use the pseudo random noise to reconstruct the disparity in uniform regions, the disparity reconstruction appears to be noisy in these areas. Once again, deactivating the refinement module removes the problem. This is not surprising since the StereoNet architecture is an hourglass hierarchical architecture. Other hierarchical methods aggregate their cost volume at different resolution (e.g. [42], [1]) before making the disparity prediction on the upsampled cost volume. StereoNet makes the prediction on a downsampled cost volume and then uses a module guided only by the initial image to upsample the disparity [14]. This approach makes StereoNet fast, but also very reliant on the appearance of the input images. This makes generalization to active stereo difficult.

# 7 Fine-tuning the models which failed to generalize

Can the problems some methods encountered when trying to generalize their predictions from passive to active stereo image pairs be eliminated by fine-tuning the aforementioned architectures on the active stereo images of our dataset? To test this, we fine-tuned ACVNet [37], Cascade-Stereo [9] and StereoNet [14] on our test set for 10 epochs. To avoid any problem of catastrophic forgetting, we ensured that the learning rate is kept low, at $5e - 5$ per mini-batch. Each mini-batch is made of four images and each epoch contained 103 mini-batches. The loss used for training the initial model was used for fine-tuning along with all hyperparameters specified for the original model.

Table 5: Evaluation of the three fine-tuned methods, ACVNet [37], Cascade-Stereo [9] and StereoNet [14].

| Method | Active stereo images | | | | | |
| --- | --- | --- | --- | --- | --- | --- |
| | $RMSE_T \downarrow$ | $MAE_T \downarrow$ | $BAD_{0.5} \downarrow$ | $BAD_1 \downarrow$ | $BAD_2 \downarrow$ | $BAD_4 \downarrow$ |
| ACVNet [37] (original) | 3.49px | 1.31px | 23% | 13% | 8% | 5% |
| ACVNet [37] (active stereo fine-tuned) | 2.16px | 0.66px | 17% | 7% | 4% | 3% |
| Cascade-Stereo [9] (original) | 4.48px | 2.07px | 64% | 33% | 14% | 8% |
| Cascade-Stereo [9] (active stereo fine-tuned) | 2.11px | 0.66px | 20% | 7% | 4% | 2% |
| StereoNet [14] (original) | 7.74px | 1.79px | 44% | 23% | 12% | 7% |
| StereoNet [14] (active stereo fine-tuned) | 7.50px | 1.78px | 56% | 29% | 13% | 7% |

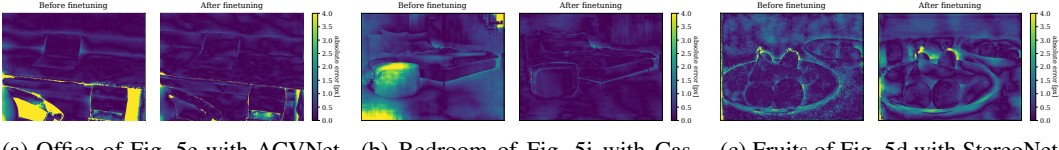

(a) Office of Fig. 5e with ACVNet [37].  (b) Bedroom of Fig. 5i with CascadeStereo [9].  (c) Fruits of Fig. 5d with StereoNet [14].

Figure 8: Error maps of ACVNet, CascadeStereo and StereoNet when fine-tuned on our dataset.

The results after fine-tuning both methods are reported in Table 5. One can observe that ACVNet and Cascade-Stereo exhibit a drastic improvement in their performance.

The visual inspection of the results reveals that for most image regions, Cascade-Stereo error is now mostly below one pixel, showing no large visible artifacts. ACVNet sees similar improvements to Cascade-Stereo. This shows that the methods were able to adapt to active stereo after fine-tuning.

For StereoNet, the RMSE improved, the $BAD_{0.5}$ and $BAD_1$ metric degraded, while the other metrics stagnated. No more artifacts correlated to the active pattern points distribution are visible. However, new edge artifacts appear in certain images. This indicates that the fine-tuning is turning the refinement module off instead of adapting it to the projected dot pattern. This, in turn, shows that an architecture largely reliant on the smoothness of the input images is ill-suited for active stereo vision. ActiveStereoNet solved this issue by decoupling the image convolutions from the disparity convolutions in their refinement module [44], which gave it more flexibility to distinguish the active pattern from real objects.

# 8 Conclusion and Perspectives

We proposed the first dataset and associated benchmark that enables the comparison of the relative performance of stereo vision algorithms when applied to active and passive stereo. Using this dataset, we undertook extensive experiments to evaluate the performance of twenty state-of-the-art end-to-end deep learning models. The reported results show that it is possible, to a certain extent, to use methods trained for passive stereo for active stereo vision. This work also shows that the weak point of those architectures is the final refinement layers. Using our training set, we were able to improve the performances of StereoNet [14] and CascadeStereo [9], which had difficulty generalizing to active stereo. StereoNet, the architecture reliant on the smoothness of the input image pair for refinement, still had very poor results, indicating that models that favor shapes prior over appearance priors are more robust.

Active stereo is an important subfield of stereo-vision. By using our proposed dataset, we were able to examine the generalization ability of current deep learning models. The dataset can also be used to fine-tune these deep learning stereo models for active stereo vision. In the future, we expect to see a growing number of ever larger deep neural networks for stereo vision. Thus, being able to evaluate their generalization ability will become more and more important, and our dataset will prove invaluable in this regard.

## Acknowledgments and Disclosure of Funding

We thank the reviewers for their helpful comments. The authors have no competing interests to disclose. This research is supported by the Australian Research Council grants ARC DP220102197 and ARC DP210101682.

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
