# OpenReview forum: "Active-Passive SimStereo - Benchmarking the Cross-Generalization Capabilities of Deep Learning-based Stereo Methods"
_NeurIPS.cc/2022/Track/Datasets_and_Benchmarks — NeurIPS 2022 Datasets and Benchmarks _

### Official Review · Reviewer_PreG · 2022-07-18
**Review of Stereo vision dataset**

**Rating:** 6
**Confidence:** 3

**Strengths:**

Introduction of a benchmark setup with different computer vision algorithm. This makes it easy to evaluate the dataset.
As it is the first rendered based dataset for stereo vision it is interesting for further synthetic data generation. It has therefore some novelty.

**Weaknesses:**

The mathematical notation of the paper and naming convention has some weaknesses. Such as using the operator for the divergence as the variance and also using a true / false statement in a sum. Also multiple operators for the product are used, which makes reading the formulas rather complicated.

Experimental analysis to the other datasets would be helpful. To better compare the benchmarks and the model performances, it would be helpful, if all models were trained on the same dataset and then evaluated on different benchmarks, then the score should be compared with the possible peak performance on the same benchmark as just such the generalisation can be better evaluated.

**Additional Feedback:**


As the authors of the paper finetuned some of the models I would assume, that GPU time was used to train those. Also rendering would have been done with the help of a GPU. Therefore I think the GPU point on the checklist is not answered correctly.

**Clarity:**

If the mathematical notations, naming conventions and missing definitions are neglected the paper was for me understandable and easy readable.


**Correctness:**

As mentioned in the weaknesses the benchmark makes for me not so much sense, as the models are all trained on different training data and are evaluated on the test data. This makes a comparison rather difficult. Also the generalisation claim is dubious, as there is no baseline establish, how well the model could fit to the data and we therefore don’t know any discreptance, nor is the test accuracy for the trained models on their data reported.

**Documentation:**

Within the supplementary material I couldn't find a documentation for the dataset.

**Ethics:**

As the images are generated by rendering, I see there no problems with the ethics


**Relation To Prior Work:**

The paper is related to previous work as stereo methods are already popular in computer vision. Their main application here is autonomous driving. The authors made it clear that their dataset is the first, which provides the active - passive component, which can be useful to specific task.
In general the authors mentioned important datasets and also described how theirs is connected to the existing ones.

**Summary And Contributions:**

The presented paper introduces a dataset for active and passive stereo computer vision methods.
Here for the first time an active passive stereo dataset has been created. The authors show that the noise introduce in the paper does not affect negatively most models. They also show a proposed benchmark for their dataset.

---

### Official Review · Reviewer_SFhb · 2022-07-22
**A novel synthetic dataset for assessing active stereo matching**

**Rating:** 5
**Confidence:** 3
**Correctness:** 1) PSMNet, StereoNet and ActiveStereo…

**Strengths:**

1) The presented ActivePassive SimStereo dataset is the pioneer dataset containing active stereo: the previous works dedicated to active stereo did not publish the experimental data.

2) During the evaluation procedure, the entire set of 10 different stereo methods are tested on the proposed benchmark. This requires a vast amount of work, after all. The intention to provide the most comprehensive picture is obvious and makes a good impression, even if the set of tested methods seems controversial.

3) The authors also report an ablation study of fine-tuning stereo methods on the ActivePassive StereoSim dataset. According to the reported metrics, the fine-tuning allows achieving a notable performance gain, which is a valuable practical result.

**Weaknesses:**

1) According to the comparison of the resolution and disparity ranges of existing datasets, the proposed ActivePassive SimStereo dataset contains much fewer images at significantly smaller resolution than UnrealStereo4K and SceneFlow. The authors claim that these simulated datasets are not realistic, while the proposed dataset created via a physically-based rendering via raytracing is realistic enough. However, this might be an overclaim, as the degree of realism is quite subjective and hardly can be measured via a formalized approach. Surely, the novel dataset is more realistic; yet, it is not convincing that reducing the domain gap between real and synthetic data can compensate the minor dataset volume and small image resolution in comparison to the existing datasets.

2) The ablation study is organized pretty badly. I would recommend comparing the performance of fine-tuned methods with the same models not being fine-tuned in the Tables. Currently, only the results of fine-tuned models are reported in this section, so one needs to return to the previous section to compare against non-fine-tuned models. Same for the visualizations: it would be more evident if there would be a comparison against the models not being fine-tuned; otherwise it is not very informative.

**Additional Feedback:**

The first active stereo benchmark seems to be a great contribution. I would definitely accept this paper if the proposed dataset contained more images at a larger resolution, and more recent methods were employed in evaluation.

**Clarity:**

The text is a bit overcompicated: I would recommend to simplify it by splitting long, hard-to-follow sentences into several parts. Nevertheless, the paper is written relatively well.

**Documentation:**

The authors provide a simulation toolkit used to produce the dataset. For evaluation, the authors select 10 methods for which the source code and the pre-trained models are available, so they can be easily employed to reproduce the results as well. However, I did not find any details of the training procedure.
As for the rest of issues, namely, the data acquisition, the dataset structure, maintenance, and licensing, the provided information is sufficient.

**Ethics:**

The synthetic dataset contains no images of people, so there should be neither privacy issues nor issues related to social groups representation.

**Relation To Prior Work:**

The authors provide quantitative comparison against existing stereo datasets featuring passive stereo, reporting image resolution and disparity range. However, the superiority over existing passive datasets is not obvious from the comparison. As stated in the paper, the proposed dataset is the first one containing active stereo, so there is not any previous works to compare with. Overall, the difference between the novel dataset and its predecessors is explicated clearly.

**Summary And Contributions:**

The paper introduces an Active-Passive SimStereo dataset. This is a synthetic dataset created with physically-based rendering engine Blender, that features 528 realistic and non-realistic images (a simulation toolkit used to produce the dataset is made available). This dataset along with the associated benchmark is intended to be used to study domain transfer between active and passive stereo vision. Alternatively, Active-Passive SimStereo dataset can be used to fine-tune the deep learning-based stereo models for active stereo vision. In this work, the authors analyze the performance of state-of-the-art trainable stereo matching methods given passive or active stereo images from the proposed dataset. The reported metrics show that existing methods can generalize to active stereo if trained on passive stereo, which is a promising result.

---

### Official Review · Reviewer_fmyK · 2022-07-27
**A benchmark and dataset to study the effectiveness of active, instead of passive stereo images.**

**Rating:** 6
**Confidence:** 4
**Clarity:** Comments concerning clarity of the pa…

**Strengths:**

**Dataset for benchmarking**:
- By providing passive and corresponding active stereo images, this paper provides a dataset that could be used to evaluate the generalization capabilities.

**Strong benchmarking and domain transfer study**:
- The authors provide valuable insights on generalization efficacy of different architectures.
- Extensive evaluation using different metrics shows that it is possible to generalize models trained on passive data to active stereo.
- Albeit not completely conclusive, the insights gained by the author surrounding the architectures such as refinement layers should be useful for the community to understand the deep neural networks more. Another study that requires further scrutiny and is not completely conclusive is the statement surrounding shape vs. appearance cues for generalization. Such studies could open the paths of research towards explainable AI systems.


**Weaknesses:**

**Sim to real quantification:**
- Although the authors show generalization ability by simply "inferring" the existing methods using the new dataset, it would be useful for the community to know whether the above dataset could be used to go from simulation to real environments.
- In other words, it would be useful to see whether the model trained on the proposed dataset generalizes well in real world domain, where the actual stereo systems would be deployed.

**Metrics and documentation:**

- The metrics are not systematically used and the reader often gets confused with different terminologies.
- Since multiple metrics are introduced, it would be better if the authors clearly state which metrics should be "higher" and which should be "lower" in Table 2 and Table 3 for easier comparison among different methods and metrics.
- Line 226: What is "problematic" methods? It was not mentioned or specified anywhere before this. Please explain terms before they are used in the sections.
- Line 232: Table 2 numbers seem counterintuitive. If these networks were trained on Passive Stereo data (meaning: no pseudo-random patterns), it is confusing as to how the inference of these models on the Active Stereo data gives less error numbers? Does that mean that these networks are not only generalizing to completely new distribution of data (Active Stereo data), but rather it is saying that these networks are actually performing even better when inferred on completely new distribution of data (Active Stereo data), compared to its original distribution (Passive Stereo data).
Except for ActiveStereoNet (which uses the active stereo data), what is it in the rest of the architectures that were trained on Passive stereo data that they are able to handle the pseudo random patterns as informative and not discarding them as noise?


**Additional Feedback:**

Please see the "Weaknesses" section.

Apart from that, a suggestion for Table 2,3: One simple trick would be to add up or down arrow indicating which numbers should be higher or lower, respectively when discussing the metrics.

**Correctness:**

Apart from some questions raised in the "Weaknesses" section, the claims made in the submission are reasonable. The dataset is constructed in a sound way and released for the public to access via a public URL.

**Documentation:**

The documentation of the dataset is clear and the detailed instructions on accessing the dataset are provided by the authors via a public URL.

**Ethics:**

The datasheet associated with the dataset reasonably answers the ethical concerns of the collected dataset.

**Relation To Prior Work:**

The relation to prior work is addressed clearly and the authors show how their specific dataset is different from existing dataset. Moreover, the authors make the motivation of generating this dataset clear which helps the reader understand how is it different from the existing works.

**Summary And Contributions:**

**Summary of contribution**:

- This paper proposes a study of domain transfer and generalization abilities of different state-of-the-art (SOTA) deep learning-based stereo architectures when presented with active, instead of passive, stereo images.
- To this end, they release a synthetic dataset. This new dataset contains active as well as passive stereo images which the authors use to benchmark different SOTA and their generalization ability.
- This paper provides some interesting conclusions regarding domain adaptation. This paper also provides insights on limiting and positive features of competing architectures using their dataset.

---

### Official Review · Reviewer_MHDH · 2022-07-28
**Not novelty enough, and the dataset has major flaws**

**Rating:** 5
**Confidence:** 4

**Strengths:**

1. The paper is fairly well written and easy to follow.
2. The experimental design is appropriate, and the motivation is clearly explained.
3. The active stereo image in Active-Passive SimStereo looks good, and the authors have also provided an implementation of the structured light simulation program, which will benefit the stereo community.

**Weaknesses:**


1. There are serious problems with the dataset, see Correctness
2. Most algorithms in the field of low-level vision have good generalization, and it is common to apply mainstream stereo matching on active images, and I have done so. This makes the paper's ideas less innovative.

**Additional Feedback:**

No.

**Clarity:**

Yes, the paper is with good format and contents.


**Correctness:**

- There are some test set data items in the training set, 13 sets in total, which are: Minicubes5, childroom_4, dining_room_1_1, hacienda4, mannequins_10, music1_7, office1, office_top4, park10, person1_8, person3_4, roundntwist4, sleeping_room_1_13
- Generation method of Sceneflow in Table 1 should be ray tracing rendering. Sceneflow was generated by Blender in 2016 when Blender only supported the ray tracing engine CYCLES.

**Documentation:**

Yes, the dataset and the simulation toolkit have clear and detailed documentation.


**Ethics:**

No.


**Relation To Prior Work:**

Yes.


**Summary And Contributions:**

This paper presents the Active-Passive SimStereo dataset, to evaluate the performance gap between passive and active stereo images for stereo matching algorithms.

The authors evaluate the generalizability of 10 stereo matching algorithms and do a detailed analysis, and find that most of the algorithms have good generalizability on active stereo images.

---

### Meta-Review · Area_Chair_iLqF · 2022-09-10

**Recommendation:** Accept
**Confidence:** 4

**Metareview:**

The final scores for this submission are borderline, but the reviewers seem to agree on the positives (the contributions of the dataset itself) while the authors have gone through great lengths to address criticisms in terms of number of algorithms compared, the size of the dataset, the ability of the dataset to generalize to the real world, and the writing quality. Reviewer MHDH was also very thorough to catch an unintended overlap of train and test data, which the authors corrected. Taken as a whole, I believe the authors have done a great job addresses the perceived weaknesses of the manuscript and dataset. I also believe it is not obvious that deep methods would generalize from passive to active stereo (a concern of reviewer MHDH), so I believe this is a worthwhile study.

---

### Decision · Program_Chairs · 2022-09-16

Accept